# Mechanical, Barrier, Antioxidant and Antimicrobial Properties of Alginate Films: Effect of Seaweed Powder and Plasma-Activated Water

**DOI:** 10.3390/molecules27238356

**Published:** 2022-11-30

**Authors:** Hege Dysjaland, Izumi Sone, Estefanía Noriega Fernández, Morten Sivertsvik, Nusrat Sharmin

**Affiliations:** 1Department of Chemistry, Bioscience and Environmental Engineering, University of Stavanger, 4021 Stavanger, Norway; 2Department of Processing Technology, Nofima AS, 4021 Stavanger, Norway; 3European Food Safety Authority, Via Carlo Magno 1A, 43126 Parma, Italy; 4Department of Food Safety and Quality, Nofima AS, Osloveien 1, 1430 Ås, Norway

**Keywords:** alginate, seaweed, plasma-activated water, antioxidant properties, antimicrobial properties

## Abstract

The incorporation of natural fillers such as seaweed may potentially enhance the properties of biopolymer films. In this study, we investigated the effect of seaweed powder as a bio-filler in alginate-based films at different concentrations (10, 30, and 50%, *w*/*w* alginate) and particle sizes (100 and 200 μm) on the mechanical, barrier, antioxidant, and antimicrobial properties of alginate which are essential for food packaging applications. Initially, mechanical properties of the alginate films prepared at different temperatures were evaluated to find the optimal temperature for preparing alginate solution. The addition of seaweed powder did not have any positive effect on the mechanical properties of the alginate films. However, the barrier (water vapor transmission rate) and antioxidant properties were improved with the addition of seaweed filler regardless of concentration. In addition, selected films were prepared in plasma-activated water (PAW). The mechanical properties (tensile strength, but not elongation at break) of the films prepared with PAW improved compared to the films prepared in distilled water, while a significant decrease was observed when incorporated with the seaweed filler. The films prepared in PAW also showed improved barrier properties compared to those prepared in distilled water. The antimicrobial activity of the alginate-seaweed film-forming solution was in general more pronounced when prepared with PAW and stored at 10 °C, particularly at the highest concentration of the film-forming solution (83.3% *v*/*v*). A more pronounced inhibitory effect was observed on the Gram-positive *S. aureus* than on the Gram-negative *E. coli*, which has been attributed to the different composition and structure of the respective cell walls. This study has demonstrated the potential of seaweed filler in combination with PAW towards enhanced functionality and bioactivity of alginate films for potential food packaging applications.

## 1. Introduction

In recent years, the use of fossil fuel-based plastic materials for food packaging has caused severe environmental concerns, such as depleting natural resources, littering, and global climate change [1,2,3]. Global plastic production has increased from 1.5 million metric tons in 1950 to 367 million metric tons in 2020 [4], and around 40% of all plastics produced are allocated to packaging. Less than a fifth of plastic is recycled globally [5], and 79% of the plastic produced is released into the natural environment [6]. When plastic is introduced into landfills, it can be transported from land to rivers and potentially end up in the ocean [7]. IUCN et al. (2021) claim that at least 14 million tons of plastic waste leak into the ocean every year, leading to the devastation of marine ecosystems [8].

A promising alternative to reduce the use of plastic materials could be natural biopolymers. Natural biopolymers that are directly extracted from biomass include proteins, polysaccharides, and lipids. They have several advantages such as a broad range of chemical composition, flexibility, durability, high gloss, clarity, high accessibility, and low cost. In addition, they are renewable, non-toxic, and biodegradable. Among different natural biopolymers, polysaccharides such as chitosan, agar, cellulose, starch, and alginate are the most promising for production of food packaging since they are not irreversibly denatured by heat, are more inexpensive, and offer better chemical stability than proteins and lipids [3,9,10,11].

Alginate is a salt of alginic acid, usually extracted from various species of brown algae, such as *Laminaria* spp. and *Macrocystis pyrifera* [12]. Alginate is a linear, water-soluble biopolymer composed of monomeric units of 1-4ß-D-mannuronic acid, called M blocks, and α-L-guluronic acid, called G blocks [13,14]. Due to its many beneficial properties, alginate is used in various industries, including food, medicine, textiles, cosmetics, and pharmaceuticals, as a thickening, film-forming, stabilizing, gel-generating, and emulsion-stabilizing agent [15,16]. Compared to other natural biopolymers alginate films have been reported to possess superior mechanical strength, flexibility, and O_2_ barrier properties. Additionally, alginate is relatively tasteless and odorless, which is beneficial for food packaging applications [3,17,18]. Alginate does not have any inherent antimicrobial properties and has very low water vapor barrier properties. Many studies focus on improving the antioxidant and antimicrobial properties of alginate films and coatings by adding different filler additions [19,20,21,22].

Seaweed is a macroalga found in almost every aquatic environment in all geographic phases [23]. Seaweed can be classified into three main groups: green algae, red algae, and brown algae. It has become one of the most promising biopolymer sources due to its chemical and physical properties and its nutritional value [23,24]. Seaweeds are considered as sources of bioactive compounds, as they produce a variety of secondary metabolites containing different biological activities [25]. Additionally, they have a high content of proteins, carbohydrates, and lipids. They contain phenolic compounds, acting as antioxidants by chelating metal ions and preventing radical formations. Thus, the high content of natural antioxidants can rapidly react with the free radicals, thereby inhibiting the extent of oxidative deterioration. In addition, these natural antioxidants can, in the same manner, also contribute to increasing the shelf life of foods [25,26]. As an organic filler, seaweed offers many advantages such as antimicrobial and antioxidant properties [25,26,27], and it is fully recyclable, cheap, and easy to obtain [28]. Norway has a long coastline of cold, temperate waters, in which several hundreds of brown, red, and green species of seaweed grows [29]. The extensive coastal areas are well suited for the cultivation of seaweed which does not require land, fresh water, or fertilizers [30,31]. However, minimal research has been performed on the addition of seaweed powder as a filler to any biopolymer solution for enhanced functional properties.

The effect of seaweed filler will depend on the species, particle size, surface activity, polymer matrix interaction, filler loading, and chemical composition. Jumaidin et al. (2017) added seaweed waste as a bio-filler to sugar palm starch/agar composites in concentrations from 0 to 40 wt.%, which improved tensile strength and elongation at break up to 30 wt.% [32]. A decrease in tensile strength and elongation at break was reported by Madera-Santana et al. (2015) [33] when seaweed waste was incorporated into polylactic acid composites (PLA) in concentrations from 5–20 wt.%, and by Bulota et al. (2016), where algae-by products mixed with diatomaceous earth was added to PLA films in concentrations of 20–40 wt.% [34]. The decrement at specific concentrations was explained by the impact of matrix disruption with high filler content, leading to a lack of stress transfer from matrix to filler [32]. In addition, reduced mechanical properties were observed when the particle size increased [34].

The application of cold plasma (CP) has been reported to enhance the functional and antimicrobial properties of biopolymers [35,36]. CP is a sustainable non-thermal disinfection and surface modification method [37]. CP consists of electrons, ions, free radicals, excited or non-excited molecules and atoms, and ultraviolet (UV) photons, which can break covalent bonds and trigger a broad spectrum of chemical reactions [38]. CP treatment of water, also known as plasma-activated water (PAW), causes changes in oxidation–reduction potential (ORP) and conductivity and results in the generation of reactive oxygen and nitrogen species (RONS) and often acidic pH (depending on the buffering capacity of the source water) [37,39]. PAW is neither classified as a chemical reagent nor a natural resource but as purified water [37]. Sharmin et al. (2021a) reported that sodium alginate films prepared with PAW showed better mechanical and water barrier properties than films made with deionized water [17]. Another study performed by Sharmin et al. (2021b) reported that the tensile strength and modulus increased when alginate films incorporated with silver nanoparticles were prepared with PAW [40]. Moreover, Okyere et al. (2022) reported that plasma technology, including PAW, could be an effective alternative for the modification of starch [36].

The aim of this study was to improve the functional, antioxidant and antimicrobial properties of alginate-based biopolymer films via the addition of seaweed powder as a bio-filler for different food packaging applications. It was hypothesized that the addition of seaweed bio-filler will increase the barrier, antioxidant and antimicrobial properties of the alginate-based films. However, it was also expected that the addition of seaweed powder might have some negative impact on the mechanical properties of the films. Therefore, alginate films were also prepared with plasma-activated water (PAW) with a hypothesis that the reactive oxygen and nitrogen species present in PAW will further increase the mechanical, barrier, antioxidant and antimicrobial properties of the seaweed-containing alginate films.

## 2. Results and Discussion

### 2.1. Effect of Temperature

#### Mechanical Properties

To observe the effect of temperature on the mechanical properties of alginate, the alginate solution was prepared at different temperatures. The solution was stirred for approximately 30 min until the alginate was completely dissolved. Figure 1a,b show the tensile strength (MPa) and elongation at break (%) of the alginate films prepared from the solution at room temperature, 35 °C, 50 °C, 70 °C, and 120 °C, respectively.

The tensile strength of the alginate film prepared at room temperature was 95.14 ± 12.96 MPa. When the temperature of the solution was increased to 35 °C, no significant change was observed in the tensile strength (92.96 ± 11.86 MPa, *p* = 0.757). However, when the temperature of the film-making solution was further increased to 50, 70 and 120 °C, the tensile strength decreased to 73.67 ± 11.40, 73.34 ± 13.42 and 71.08 ± 15.85 MPa, (*p* = 0.012, *p* = 0.033 and *p* = 0.022), respectively. No significant difference was observed between the elongation at break of the alginate films prepared at different temperatures. The elongation at break for the alginate film prepared at room temperature was 3.70 ± 0.85%. For the films prepared at 35, 50, 70 and 120 °C the elongation at break was 4.42 ± 1.59%, 4.51 ± 0.85%, 2.94 ± 0.81% and 3.80 ± 0.99% (*p* = 0.349, *p* = 0.131, *p* = 0.190 and *p* = 0.873), respectively.

As mentioned earlier, the effect of temperature on alginate was performed to identify the most suitable temperature to make alginate solution without hampering the mechanical properties. In relation to the presented results, it was concluded to perform the rest of the experiments at room temperature. As for the knowledge of the author, limited information is to be found about the influence of temperature during the preparation of the alginate solution. However, as could be observed from Figure 1, the tensile strength significantly decreased when the temperature of the film-making solution was 50 °C or higher, which could be due to the depolymerization of the alginate structure at high temperatures. Mao et al. (2011) studied the depolymerization of sodium alginate by oxidative degradation and reported that the oxidative depolymerization process was accelerated by increased temperature [41]. In addition, they investigated the relationship between alginate molecular weight and reaction temperature up to 80 °C and reported that temperatures above 40 °C lead to considerably enhanced depolymerization of alginate [41], which may explain the decreased tensile strength at higher preparation temperatures of the alginate films in the present study.

### 2.2. Effect of Seaweed Concentration and Particle Size

#### 2.2.1. Mechanical Properties

Figure 2 and Figure 3 show the tensile strength (MPa) and elongation at break (%) of alginate film prepared with seaweed filler within a particle size range of 200 and 100 µm in three different filler concentrations (10, 30 and 50% (*w*/*w*)).

Figure 2a shows that the tensile strength of the alginate film decreased significantly with seaweed filler of 100 µm at all concentrations. The pure alginate film had a mean tensile strength of 95.14 ± 12.96 MPa, which decreased to 56.47 ± 6.83, 51.54 ± 3.81, and 46.27 ± 4.59 MPa with 10, 30, and 50% seaweed filler (*p* < 0.001), respectively. Although the tensile strength of the seaweed-containing samples decreased with increasing filler addition, no significant difference was observed between the 10 and 30% seaweed-containing films for 100 µm. Similar to 100 µm, seaweed filler of 200 µm (Figure 2b) had a significant effect on the tensile strength of the alginate film at all concentrations. The tensile strength significantly decreased to 48.58 ± 7.29, 41.85 ± 6.16, and 40.11 ± 3.63 MPa, with 10, 30 and 50% seaweed filler (*p* < 0.001), respectively. Although the tensile strength slightly decreased with increasing concentration of seaweed, the post hoc test indicated that there was no significant difference in tensile strength between the alginate films with different seaweed concentrations. The tensile strength of the alginate film decreased when the particle size range increased from 100 to 200 µm. With 10% seaweed filler, the tensile strength was lower by 13% for alginate film with seaweed filler of particle size 200 µm compared to the film with 100 µm filler (*p* = 0.082). However, despite a relatively large decrease in tensile strength with increased particle size, the difference was not significant, possibly because of large standard deviations. For 30% and 50% seaweed-containing films, the tensile strength was significantly lower by 18 and 13% (*p* = 0.008 and *p* = 0.028), respectively, for films containing 200 µm filler as compared to 100 µm films.

A similar decreasing trend with increasing seaweed concentration was also observed for the elongation at break of the alginate film (Figure 3). As can be seen from Figure 3a (100 µm), a significant effect on the elongation at break was observed for all seaweed concentrations. For pure alginate film, the elongation at break was 3.70 ± 0.85%. After the addition of 10, 30 and 50% seaweed filler, the elongation at break decreased to 2.53 ± 0.27%, 2.79 ± 0.30%, and 2.37 ± 0.23% (*p* = 0.009, *p* = 0.032 and *p* = 0.004), respectively. In addition, the elongation at break of 50% seaweed-containing film was significantly lower than the 30% seaweed-containing film. For seaweed of 200 µm, the decrease in elongation at break was only significant with 30 and 50% seaweed filler (Figure 3b). With the addition of 10% seaweed, the elongation at break decreased to 3.10 ± 0.50% (*p* = 0.162). With 30 and 50% seaweed addition, the elongation at break further decreased to 2.80 ± 1.20% and 2.46 ± 0.30% (*p* = 0.029 and *p* = 0.007), respectively. In addition, the post hoc test showed a significant difference in the elongation at break between 10 and 50% seaweed-containing samples. Moreover, no significant difference was observed in the elongation at break for 30 and 50% seaweed filler (*p* = 0.954 and *p* = 0.586) for the different particle sizes. However, a significant difference with small effect sizes was observed with 10% seaweed filler, where the elongation at break was lower by 18% for seaweed-containing films of 100 µm as compared to that of 200 µm (*p* = 0.034).

A number of studies have reported the effect of different filler additions on the mechanical properties of biopolymer films, some of which are in line with the result of the present study, while some are not. Some studies have reported general decreases in mechanical properties, where seaweed of different particle sizes and concentrations was added to composites other than alginate [33,34]. Madera-Santana et al. (2015) [33] added seaweed waste in concentrations from 5–20 wt.%, and Bulota et al. (2016) [34] added algae by-products mixed with diatomaceous earth in concentrations from 20–40 wt.% to PLA films. They both reported a decrease in tensile strength and elongation at break, which is consistent with the present study, where the mechanical properties significantly decreased (*p* < 0.050), as previously mentioned. Decrements in elongation at break were explained by increased stiffness, leading to less flexible and more brittle films with added filler [33], which could be the case for the addition of seaweed powder to alginate films. They also explained that the tensile strength decreased as the filler content increased, disregarding the type of filler and particle size. However, similar to this study, the strength decreased more with bigger particle sizes, which could be explained by the fact that smaller particles are better distributed within the matrix as compared to bigger particles [34]. A similar observation on particle size was observed when Riahi et al. (2022) incorporated sulfur quantum dots (SQDs), nanosulfur (SNP), and elemental sulfur (ES) into alginate composite films [42]. It was reported that the tensile strength of the film incorporated with SQDs increased by about 18% as compared to the control alginate film caused by good compatibility between the filler and the polymer. However, for films incorporated with SNP and ES, the tensile strength decreased by 29 and 14%, respectively, due to their larger particle sizes, causing a non-uniform dispersion and aggregation in the film, in contrast to the evenly distributed SQD film with smaller particle size [42].

Ideally, a filler should act as a discontinuous phase within the composite, enhancing the properties distributed within the matrix. Then, both filler and matrix usually complement each other and lead to a composite with better functional properties [9]. However, this was not the case for the mechanical properties of alginate composite films when seaweed powder was added as a filler and could be explained by the seaweed not becoming a part of the alginate structure [9].

#### 2.2.2. Barrier Properties

Figure 4 shows the effect of the addition of seaweed filler at different concentrations (10 and 30% (*w*/*w*)) on the WVTR of the alginate films. The size of the seaweed filler was within the size range of 100 µm. Films prepared with seaweed filler of 200 µm were too brittle to perform the experiment.

The pure alginate film had a WVTR of 136.89 ± 8.24 g·m^−2^·h^−1^. With 10% seaweed filler, the WVTR significantly decreased to 123.05 ± 4.97 g·m^−2.^h^−1^ (*p* = 0.028). A significant effect was also seen at 30% seaweed concentration, where the WVTR further decreased to 112.86 ± 2.79 g·m^−2^·h^−1^ (*p* = 0.001). At 50% seaweed filler, the films were too brittle to handle, and therefore the experiment was not possible to implement. Moreover, a significant difference in the WVTR was observed between the 10 and 30% seaweed-containing films (*p* < 0.001).

To avoid large amounts of moisture transferred from the atmosphere to the food, good water barrier properties of the packaging films are crucial. As mentioned previously, the water barrier properties of polysaccharide-based biopolymers are usually poor because of their hydrophilic nature, causing them to absorb water [9]. However, it has been reported that the incorporation of different fillers, such as nanocellulose, eggshell powder, etc. [43,44,45], can lead to improved water barrier properties since the filler might interfere with the hydrophilic portion of the film, thus reducing its hydrophilicity. Huq et al. (2012) [44] and Abdollahi et al. (2013) [45] reported that the water vapor permeability of alginate films decreased by 31% and 18% with increasing filler loading of NCC and CNP, respectively. They reported that the improved water vapor permeability was due to the increased tortuous path in the films, leading to slower diffusion mechanisms and thus decreased permeability [44,45]. They also explained that the water barrier properties can improve when the filler is less permeable to water than the matrix and is homogeneously dispersed within the matrix [44,45]. Similar observations were reported by Jiang et al. (2018), who explained that the addition of eggshell powder led to significant improvements in the water vapor permeability of corn starch films, due to an increased curved path, reduced hydrophilicity and increased density of the corn starch film with filler [46]. These findings are in line with the present study, in which a significant reduction in the WVTR of alginate films was seen when seaweed filler of 100 µm was added in concentrations of 10 and 30% (*p* < 0.050). In addition, the WVTR further decreased with increased concentration of filler (*p* < 0.001). Moreover, similar observations were made when sugar palm-derived cellulose was reinforced to sugar palm starch biocomposites [47] and when eggshell powder was incorporated into Polyvinyl alcohol biocomposites [48].

#### 2.2.3. Antioxidant Properties

The mean DPPH scavenging activity of alginate film with filler at 10, 30, and 50% (*w/w*) is presented in Figure 5a with a particle size of 100 µm and 200 µm (Figure 5b) in four different concentrations of alginate (0.5, 1.0, 2.0 and 3.0 mg/mL).

Increased DPPH scavenging activities were observed with increasing filler and alginate concentrations, while no concentration dependence in the scavenging activity was found in alginate film without filler. Similar trends were followed regardless of particle size. The observed effect on the antioxidative activity implied the involvement of seaweed particles and/or bioactive compounds released into the film-forming solutions. The latter was confirmed with alginate film containing seaweed-extract solutions which exhibited higher DPPH scavenging activity than that of pristine film at increasing alginate concentrations. Potent antioxidant activities of seaweed extract have been well documented within film matrix attributed to redox capacity of extracted phenolic compounds [27,49]. This study demonstrated their antioxidative effect as seaweed filler within the alginate-based film matrix, exhibiting concentration-dependent behavior similar to those of other bioactive compounds [49]. Interestingly, when seaweed powder was incorporated directly into the film-forming solution as filler, the DPPH scavenging activity was approximately 20% higher than that of the film with seaweed extract (compared at 50% filler concentration, 25 °C), which suggested that the elevated antioxidative activities also involved the seaweed particles. The slightly (up to 6.5%) higher scavenging activities observed with filler of smaller particles may be related to film structural properties such as higher crystallinity [34], interfacial bonding and compositional uniformity of composites. Lower scavenging activity of the pristine alginate films was observed in this study than those reported in the literature. This could also be related to their structural properties, e.g., molecular weight and G/M ratio that influences the availability of hydroxyl groups and ability to donate H-atoms [12,50,51] due to species difference, growth conditions, geographical location, harvest season [51] and processing conditions [52,53], which may be compensated by seaweed filler addition as this study has demonstrated.

### 2.3. Effect of Plasma Activated Water

#### 2.3.1. Mechanical Properties

Figure 6 compares the tensile strength (MPa) and elongation at break (%), between the films prepared with distilled water and films prepared with PAW.

For the pure alginate film prepared in PAW, the tensile strength was 147.49 ± 16.74 MPa, a significant improvement of 55% as compared to the film prepared with distilled water (*p* < 0.001). With the addition of seaweed filler, the tensile strength decreased for all concentrations. The tensile strength for 10% seaweed-containing film prepared in PAW was higher by 41% (79.59 ± 10.61 MPa) compared to the films prepared in water (*p* = 0.001). For 30 and 50% seaweed filler, the tensile strength for the film prepared in PAW was 54.11 ± 9.61 and 44.35 ± 6.60 MPa (*p* = 0.557 and *p* = 0.572), respectively. Thus, at 30 and 50% seaweed filler, no significant change in tensile strength was observed for the films prepared in PAW, which possibly could indicate that the increased concentration of seaweed filler inhibited the effect of PAW. 

No significant effect on the elongation at break was seen between the samples prepared in distilled water and the samples prepared in PAW for pure alginate film and after 10% seaweed addition. However, a significant effect was observed for the alginate film with 30 and 50% filler. As mentioned previously, the elongation at break for the pure alginate film prepared in water was 3.70 ± 0.85%. For the pure film prepared in PAW, the elongation at break was 3.20 ± 0.79% (*p* = 0.342), and with 10% filler it was 2.36 ± 0.28% (*p* = 0.320). For 30 and 50% seaweed filler, the elongation at break was 1.75 ± 0.31% and 1.29 ± 0.25% for the film prepared in PAW, a significant decrement of 37 and 45% (*p* < 0.001), respectively, as compared to the film prepared in distilled water.

Sharmin et al. (2021a) studied the effect of citric acid and PAW on the functional properties of alginate and reported that the tensile strength, tensile modulus, and elongation at break of the alginate film prepared in PAW improved by 33%, 17%, and 26%, respectively, compared to the control alginate film [17]. An increase in tensile strength and modulus was also reported when alginate films were prepared with PAW and silver nitrate, which increased by approximately 33% and 46%, respectively, compared to the control alginate film [40]. Similar results were observed in the present study, where the tensile strength and modulus significantly improved by 55% and 76%, respectively, for the pure alginate film prepared in PAW compared to the alginate film prepared in distilled water. The improved mechanical properties of alginate film prepared in PAW were explained by the crosslinking of alginate chains with the nitrites and nitrates in PAW. Additionally, the RONS generated in PAW results in low pH, which leads to protonation of COO^−^ to COOH in the alginate structure, resulting in hydrogen bond formations within the alginate chains and improved mechanical properties [17,37,40]. However, the tensile strength of the film significantly decreased with the addition of seaweed powder. At 30 and 50% seaweed concentration, there was no longer a significant difference between the film prepared with PAW and the film prepared with distilled water, which may be because of the poor interaction between the alginate matrix and the seaweed powder, which overran the positive effect of PAW. Additionally, the elongation at break was lower for the films prepared with PAW, similar to what was observed when alginate silver nanocomposite films were prepared with PAW, which led to a slight decrease in the elongation at break, with no significant effect [40].

#### 2.3.2. Barrier Properties

Figure 7 compares the WVTR of alginate films with seaweed filler (10 and 30% (*w/w*)) prepared in distilled water and PAW. For the pure alginate films, the film prepared in PAW had a significantly lower WVTR than the film prepared in distilled water. The film without filler prepared in PAW had a WVTR of 119.41 ± 4.6 g·m^−2^·h^−1^, a decrement of 13% compared to the film prepared in distilled water (*p* = 0.005). A significantly lower WVTR for the film prepared in PAW could also be seen with 10 and 30% seaweed filler, where the WVTR was 102.52 ± 2.43 and 96.70 ± 5.21 g·m^−2^·h−1, corresponding to a decrease of 16 and 14% (*p* < 0.001 and *p* = 0.001), respectively, as compared to the film prepared in distilled water.

The WVTR was significantly improved for alginate films prepared in PAW (0, 10, and 30% seaweed filler) as compared to the film prepared in distilled water. As previously reported, with 50% seaweed filler, the film was too brittle to perform the experiment. Sharmin et al. (2021a) reported similar improvements in barrier properties where the alginate film prepared with PAW were significantly lower than the alginate film prepared with deionized water [17]. This was explained by the crosslinking of alginate with the reactive species in the PAW [17]. The crosslinking effect of PAW combined with seaweed powder led to further improvements in the WVTR, which could be attributed to the reduced hydrophilicity of the films and to the slower diffusion mechanism due to an increased tortuous path caused by the seaweed powder, leading to further improvements as the filler loading increased from 0 to 10 and 30% seaweed (*w*/*w*) [17,44,45].

#### 2.3.3. Antioxidant Properties

The mean DPPH scavenging activity of alginate film with filler (particle size 100 µm) at 10, 30, and 50% (*w*/*w*) with and without PAW is presented in Figure 8 in four different concentrations of alginate (0.5, 1.0, 2.0 and 3.0 mg/mL).

The DPPH scavenging activity of alginate film decreased with PAW at high alginate concentrations (at 2–3 mg/mL) as compared to film prepared in distilled water, which may be due to the oxidizing capacity of PAW [40]. The filler effect of PAW on the DPPH scavenging activity varied depending on filler and alginate concentrations, but the difference attributed to PAW addition was mostly at a magnitude of <6 % on average, except for at 0.5 mg/mL alginate with 50% filler concentration leading to approximately 10% decrease in the scavenging activity upon PAW incorporation. Similar increasing trends in the activity in relation to filler and alginate concentrations were observed, as described above. Xiang et al. (2019) reported that PAW treatment had no adverse impact on the antioxidant activity and the total phenolic compounds of mung bean sprout extracts [54]. Similarly, Xu et al. (2016) reported no significant change in the antioxidant activity of the button mushroom *Agaricus bisporus* when it was treated with PAW [55]. These results are in line with those of the present study, suggesting that PAW may not impact the antioxidant properties of the alginate films.

### 2.4. Antimicrobial Properties

The antimicrobial properties of alginate prepared with 50% (*w*/*w*) seaweed film-forming solution, alone or in combination with PAW, were investigated on the Gram-negative *E. coli* and Gram-positive *S. aureus*. The effect of temperature abuse (37 °C) and refrigeration (10 °C) conditions were investigated, as well as the effect of the concentration of the film-forming solution, i.e., 33.3 or 83.3% (*v*/*v*).

Figure 9 and Figure 10 show the concentration of *E. coli* and *S. aureus*, respectively, in alginate samples with 50% *w*/*w* seaweed prepared with either PAW or distilled water after 24 h at 37 °C and 300 rpm. The effect of the concentration of the film-forming solution (33.3 or 83.3% *v*/*v*) was assessed only on PAW-treated samples. The average initial concentration for all conditions tested was 1.34 × 107 ± 6.20 × 106 CFU/mL for *E. coli* and 1.01 × 107 ± 5.36 × 106 CFU/mL for *S. aureus*. After incubation at 37 °C for 24 h, a significant increase in viable counts (about 2 log10) was observed for the control samples (TBS) of both microorganisms, as compared to their respective initial concentrations (*p* < 0.001).

With regard to the samples containing 33.3% *v*/*v* of alginate with 50% *w*/*w* seaweeds prepared with PAW, no significant differences (*p* = 0.075) were observed in the viable counts of *E. coli* as compared to the control samples after 24 h at 37 °C. For *S. aureus*, a significantly lower concentration was observed for the treatment solution as compared to the control (*p* = 0.001), after 24 h at 37 °C, although this variation represented less than 1.0 log10 reduction. However, when increasing the concentration of the film-forming solution (alginate with 50% *w/w* seaweeds prepared with PAW) to 83.3% *v*/*v*, a significant decrease in viable counts (2.0 log10 and 2.8 log10 reductions for *E. coli* and *S. aureus*, respectively) was observed for both microorganisms after 24 h at 37 °C, in relation to the control samples (*p* < 0.001).

The antimicrobial activity of samples containing 83.3% *v*/*v* of alginate with 50% (*w*/*w*) seaweed in distilled water was also investigated. For *E. coli*, no significant difference in viable counts (*p* = 0.409) was observed as compared to the respective control in TSB after 24 h at 37 °C. However, a significant decrease was observed for *S. aureus* (1.2 log10 reductions), in relation to the control (*p* = 0.006). When comparing the samples prepared with PAW and distilled water, significantly lower viable counts in the samples prepared with PAW (*p* < 0.001) were observed for both microorganisms.

Figure 11 and Figure 12 show the concentration of *E. coli* and *S. aureus*, respectively, in the alginate samples with 50% (*w*/*w*) seaweed prepared with either PAW or distilled water after a 5- and 10-day incubation at 10 °C and 70 rpm. Similar to Figure 9 and Figure 10, the effect of the concentration of the film-forming solution (33.3 or 83.3% *v*/*v*) was assessed only for PAW-treated samples. The average initial concentration for all conditions tested was 1.73 × 107 ± 4.41 × 106 CFU/mL for *E. coli* and 7.22 × 106 ± 5.07 × 106 CFU/mL for *S. aureus*. For both microorganisms, significantly higher viable counts in the control samples (TSB) (*p* ≤ 0.008) were observed after the 5- or 10-day incubation at 10 °C, with regard to their respective initial concentrations. However, the concentration of both microorganisms remained relatively unchanged (*p* > 0.5), when comparing both incubation periods at 10 °C.

When considering the samples prepared with 33.3% *v*/*v* of alginate with 50% *w/w* seaweeds prepared with PAW, no significant differences in the viable counts of *E. coli* (*p* = 0.054) were observed after 5 days at 10 °C, as compared to the control samples. After 10 days of incubation at 10 °C, the concentration of *E. coli* marginally increased as compared to day 5 (*p* = 0.031). However, no significant change (*p* = 0.101) was still observed in relation to the control samples. For *S. aureus*, on the other hand, a significant effect was noticed both after 5 and 10 days at 10 °C (*p* ≤ 0.001) as compared to the respective control samples (≈ 1 log10 reduction). Marginal differences (*p* = 0.041) were also observed in the viable counts of *S. aureus* in the treated samples between days 5 and 10 of incubation at 10 °C.

When increasing the concentration of the film-forming solution (alginate with 50% *w/w* seaweeds prepared with PAW) to 83.3% *v*/*v*, the viable counts of *E. coli* significantly decreased after 5 and 10 days at 10 °C (1.9 and 1.5 log10 reductions, respectively), as compared to the respective controls (*p* < 0.001). Significant differences were also observed when comparing the viable counts achieved on days 5 and 10 (*p* = 0.033). Moreover, a significant decrease in viable counts (*p* ≤ 0.006) was observed when comparing the highest concentration of the test item (83.3% *v*/*v*) with the lowest one (33.3% *v*/*v*) for both 5 and 10 days of incubation at 10 °C (1.6 and 1.7 log10 reductions, respectively). With regard to *S. aureus*, a significant decrease in viable counts, as compared to control samples, was observed, after 5 and 10 days of incubation at 10 °C (1.1 and 1.3 log10 reductions, respectively). However, significant differences in viable counts were not observed, regardless of the concentration of the film-forming solution (*p* > 0.100) or the incubation period at 10 °C (*p* = 0.378).

The antimicrobial activity of samples containing 83.3% *v*/*v* alginate with 50% *w*/*w* seaweeds in distilled water was also investigated after 5 and 10 days at 10 °C and 70 rpm. For *E. coli*, the concentration at day 5 significantly decreased (0.9 log10 reduction), as compared to the control (*p* = 0.009). A marginal decrease (*p* = 0.026) was also noticed after 10 days, as compared to control samples. However, no significant differences (*p* = 0.278) were observed when comparing viable counts of *E. coli* after days 5 and 10. For *S. aureus*, the concentration significantly decreased after 5 and 10 days at 10 °C (0.7 and 0.9 log10 reductions, respectively), as compared to the respective controls (*p* ≤ 0.005). Moreover, a significant decrease (*p* = 0.020) was also observed when comparing the bacterial concentrations on day 5 and day 10. 

When comparing the viable counts of *E. coli* and *S. aureus* in samples containing 83.3% *v*/*v* alginate with 50% (*w*/*w*) seaweeds prepared with either distilled water or PAW, a significant effect of PAW was observed after both 5 and 10 days for *E. coli* (*p* = 0.020 and *p* = 0.017) and *S. aureus* (*p* = 0.030 and *p* = 0.017), respectively.

For *E. coli*, the inhibitory activity of seaweed bio-fillers was confirmed at 37 °C in combination with PAW, and at 10 °C, regardless of PAW treatment, although in both cases, only at the highest concentration of the film-forming solution. For *S. aureus*, the inhibitory activity of seaweed bio-fillers was confirmed at both 37 and 10 °C, regardless of PAW and the concentration of the film-forming solution. Overall, lower viable counts of both microorganisms were observed when the alginate–seaweed solution was prepared with PAW and incubated at 10 °C rather than 37 °C, although more pronounced log10 reductions were noticed at 37 °C, in relation to the controls in TSB. Moreover, stronger inhibitory activity was typically observed for *S. aureus*, and a more pronounced antimicrobial effect was similarly observed on both microorganisms at the highest concentration of the film-forming solution.

The antimicrobial activity observed in the present study could be attributed to the presence of seaweeds and PAW in the film-forming solutions, since sodium alginate has been reported in the literature not to cause any inhibitory effect on a broad spectrum of microorganisms. For instance, Benavides et al. (2011) incorporated oregano essential oil into alginate films and reported that the alginate control film did not inhibit the growth of pathogenic bacteria. Similarly, Han et al. (2016) reported that sodium alginate and carboxymethyl cellulose films showed no antibacterial properties in the absence of pyrogallic acid [56]. Although there is limited research on the antimicrobial activity of alginate films in combination with PAW and seaweeds, their separate inhibitory effect has been widely studied. The antimicrobial activity of PAW has been explained by the presence of RONS, high oxidation-reduction potential, and often acidic pH [57,58]. RONS affect the membrane integrity and penetrate through the cell membrane, where they interact with intracellular components, such as DNA, proteins, and lipids, and induce oxidative stress on the cell, leading to protein and DNA damage and eventually cell death [57,58]. Xu et al. (2020) reported considerable inhibitory effects on *S. aureus* biofilms with increased PAW exposure [58]. Similarly, Ma et al. (2015) reported that PAW was used to inactivate *S. aureus* on strawberries, leading to no visual spoilage on the PAW-treated strawberries after 6 days of storage [59]. In a study performed by Sharmin et al. (2021b) [40], sodium alginate-silver nanocomposites films were synthesized within PAW, containing 1-, 3- or 5 mm silver nitrate, which led to significant inhibition of the growth of *S. aureus* and *E. coli* attributed to the silver nanoparticles and irrespective of the silver nitrate concentration and test dose (3.3 or 33.3% *v*/*v*). However, PAW-treated nanocomposites, particularly at high silver nitrate concentrations and test doses, showed slightly better antimicrobial properties [40]. On the other hand, seaweed is rich in active phenolic compounds, such as phlorotannins [60]. Gupta et al. (2011) proposed that the phenolic compounds may act as a source of natural antimicrobials by attacking the cell membrane of bacteria, leading to inhibition by disruption of membrane functions such as protein and nucleic acid synthesis, enzyme activity, and nutrient uptake [26].

As previously mentioned, a more pronounced inactivation efficacy was observed on the Gram-positive *S. aureus* than on the Gram-negative *E. coli*, which could be attributed to the differences in the cell wall and membrane [26]. Gram-negative bacteria are surrounded by a thin peptidoglycan cell wall, which is surrounded by an outer membrane consisting of lipopolysaccharides. On the other hand, Gram-positive bacteria lack an outer membrane but are surrounded by a much thicker layer of peptidoglycan as compared to Gram-negative ones [40,61]. Other studies have also reported a higher antimicrobial effect of seaweeds against Gram-positive bacteria as compared to Gram-negative ones [62,63,64]. Rhimou et al. (2010) studied the antimicrobial effect of 26 marine extracts from Rhodophyceae and reported that 25 species were active against at least one of five microorganisms. Only five of the species had antimicrobial effects against *E. coli*, while *S. aureus* was reported to be the most susceptible microorganism, with 25 species showing inhibitory effects [62]. Similarly, Val et al. (2001) reported that the antibacterial activity of red, green, and brown macroalgae was less common in Gram-negative bacteria [63]. Moreover, differences between the optimum growth temperatures for *E. coli* (37 °C) and *S. aureus* (40–45 °C) and their minimum growth temperatures (4.0 and 7.0 °C, respectively), may have also contributed to the different inhibitory effects observed when comparing both microorganisms.

## 3. Materials and Methods

### 3.1. Materials

Sodium alginate (alginic acid sodium salt from brown algae, M/G ratio: 2.12) with low viscosity and glycerol (1,2,3-Propanetriol, C3H8O3) were supplied from Sigma-Aldrich (Darmstadt, Germany). DPPH (2,2-Diphenyl-1-picrohydrazyl free radical, 95%) was purchased from Alfa Aesar (Kandel, Germany). Supplied from Antibac (Asker, Norway) was 96% ethanol. Tryptone Soya Broth (TSB) and Mueller-Hinton Agar (MHA) was purchased from Oxoid (Basingstoke, UK), and Sodium Chloride (NaCl, analysis grade) and Plate Count Agar (PCA) were purchased from Merck (Darmstadt, Germany).

The seaweed (Laminaria hyperborea) with a water content of 6.3% was supplied by Dolmøy House of Seafoods (Frøya, Norway) and harvested in May 2020. After removing the stipe and holdfast, the square-shaped samples were stored at 4 °C. Nofima performed the milling of the seaweed. Briefly, the procedure involved taking frozen seaweed, thawing it, and then removing the stems. The leaves were then finely milled using a Comitrol 1700 (Urschel laboratories, Chesterton, IN, USA) with cutting heads 3K-025040U (1.016 mm opening) and 3K-010010 (0.354 mm opening). After freezing the leaves overnight at −20 °C, they were lyophilized for 48 h in a GAMMA 1-16 LSC dryer (MARTIN CHRIST GmbH, Osterode, Germany). The dried leaves were then conditioned against air at an ambient temperature for two days before freezing again at −80 °C. Finally, the frozen leaves were ground on a Retch ZM-1 Centrifugal mill (Retsch GmbH, Haan, Germany) using a ring sieve with an aperture of 0.5 mm.

### 3.2. Film Preparation

The alginate solution was prepared by adding 2 g of alginate powder to 100 mL of distilled water. The solution was then stirred using a magnetic stirrer (MR Hei-Tec, Heidolph Instruments, Schwabach, Germany) at 550 rpm for approximately 40 min at room temperature till the powder was completely dissolved. Then, the pH of the solution was measured with a FiveEasy Plus pH meter (Mettler Toledo, Columbus, OH, USA) equipped with a LE410 electrode before 20 mL of the solution was poured into 90 mm diameter polystyrene Petri dishes and left to dry at room temperature for approximately 72 h.

### 3.3. Seaweed Addition

Seaweed powder was added to the alginate solution to observe the potential effect on the functional properties of the films. At first, the seaweed powder was sieved with a 200 µm and a 100 µm sieve. The powder was then added to the alginate solution in concentrations of 10, 30, and 50% of the dry weight of alginate, stirred for approximately 30 min, and the films were casted, as described in Section 2.2.

Initially, the films were casted without glycerol. However, due to high brittleness, it was difficult to handle and test the 30 and 50% seaweed-containing films (with the particle size range 200 µm). With the 100 µm seaweed powder, a similar problem happened but only with 50% seaweed-containing films. Hence, it was impossible to perform the barrier properties of these concentrations. Therefore, 10% glycerol (*w*/*w*) was added to the alginate solution to reduce the brittleness of the films, making it possible to analyze them.

### 3.4. Mechanical Properties

Mechanical properties are determined by the evaluation of stability and resistance of the composite film to sustain the exertion of load [9]. Mechanical properties such as tensile strength, tensile modulus, and elongation at break of the films were measured using a TA. XT Plus Texture Analyzer (Stable Micro Systems, Godalming, UK). The texture analyzer was equipped with tensile grips and a 50 kg loading cell. A span distance of 25 mm and a speed of 0.85 mm/s were used for the experiment. The films were cut to a dimension of 60 mm × 15 mm (length × width). The thickness of the films varied from sample to sample due to different filler concentrations and was therefore measured with a Limit Digital Caliper. At least three replicates of each sample were cut out and analyzed using the software Exponent (version 6.1.16.0).

### 3.5. Antioxidant Activity

The DPPH scavenging activity was measured using the method by Blois MS et al. (1958) [65]. Briefly, 0.15 mM DPPH solution was prepared in 96 % ethanol upon stirring for 15 min at room temperature and stored for 1 h covered in foil to avoid exposure to light. A sample of alginate film with and without filler was dissolved in distilled water to the initial concentration of 4 mg/mL alginate. After stirring at room temperature for up to 10 min, the sample solution was further diluted to 0.5, 1.0, 2.0, and 3.0 mg/mL alginate using distilled water. Each diluted sample was mixed with equal part of DPPH solution (*v*/*v*), vortexed for 10 s and placed in the dark for 1 h. A control with DPPH solution and blank with distilled water were also prepared. The Sample solutions containing filler were centrifuged for 1 min at 13,000 rpm (Eppendorf Minispin, Hamburg, Germany) prior to absorbance measurement. The absorbance was read at 517 nm in triplicate using a multi-mode spectrometer (Synergi H1 reader with Gen5 software, BioTek Instruments, Winooski, VT, USA) and the DPPH scavenging activity (%) was calculated as follows:$$\mathrm{DPPH}\;\mathrm{Scavenging}\;\mathrm{activity}\;\left(\%\right)=\frac{A_{\mathrm{control}}-A_{\mathrm{sample}}}{A_{\mathrm{control}}}\times 100$$
where A_control_ is the absorbance of the control and A_sample_ is the absorbance of the sample.

### 3.6. Barrier Properties

To avoid large amounts of moisture transfer from the atmosphere to the food, the water barrier properties of the films are crucial, especially in food packaging applications [9]. The water vapor transmission rate (WVTR) of the films was determined according to the method described by Sarwar et al. (2018) [66] with slight modifications. Alginate films were cut into appropriate sizes and placed on top of a cylindrical tube with a diameter of 13.5 mm containing 10 mL of distilled water. The films were sealed with parafilm. The tubes were weighed using Fisher Scientific Precision Series balance (Thermo Fisher Scientific, Waltham, MA, USA), placed in an oven at 45 °C for 24 h, and then weighed again to determine the moisture permeability of the films. 

The water vapor transmission rate (WVTR) was calculated using the following equation:$$\mathrm{WVTR}=\frac{\left(w_{i}-w_{t}\right)}{A\;\times \;T}\times 10^{6}{\mathrm{gm}}^{-2}h^{-1\;}$$
where A represents the area of the bottle, T is 24 h, w_i_ is the weight of the bottle at time zero, and w_t_ is the weight after 24 h.

### 3.7. PAW Production

PAW was produced according to the method described by Risa Vaka et al. (2019) and Sharmin et al. (2021a) with slight modifications in order to ensure the highest levels of reactive oxygen and nitrogen species (RONS) and stability during storage. Briefly, the cold plasma reactor used for PAW generation was a surface barrier discharge (SBD) system with the powered and ground electrodes separated by a 1 mm thick quartz disk (144 cm^2^ discharge area) and coupled to the lid of the treatment chamber (176 × 174 × 48 mm). For a 100 mL treatment volume (3.2 mm water column), a 44.8 mm gap distance was measured between the liquid surface and the electrode. PAW was generated using an 18 kHz sinusoidal frequency, 30 min activation time, and 36 W plasma power. The system operated at atmospheric pressure, with room air as the plasma-inducing gas. The produced PAW (pH 2.3) was stored at 4 °C and used within 48 h.

### 3.8. Antimicrobial Studies

The antimicrobial properties of the selected film-forming solutions (“PAW + alginate + seaweeds” and “alginate + seaweeds”) were investigated, as described by Sharmin et al. (2021b) [40] with slight modifications. The antimicrobial studies were conducted with *Escherichia coli* CCUG 10979 (Gram-negative) and *Staphylococcus aureus* CCUG 1828 (Gram-positive). Both strains were stored at −80 °C using MicrobankTM porous beads as carriers (Microbank, Pro-lab Diagnostics, Richmond Hill, ON, Canada). The effects of temperature (37 or 10 °C) and test item concentration (33.3% *v*/*v* or 83.3% *v*/*v*) on the antimicrobial properties of the selected film-forming solutions were investigated.

MicrobankTM beads of *E. coli* and *S. aureus* were streaked onto Plate Count Agar (PCA) plates and incubated at 37 °C for 24 h (Sanyo MIR-154 refrigerated incubator, SANYO Commercial Solutions, Kennesaw, GA, USA). Then, a single colony was transferred to a 15 mL Falcon tube (Sarstedt, Nümbrecht, Germany) containing 5 mL of Tryptic Soy Broth (TSB) and incubated at 37 °C. After 24 h, appropriate serial decimal dilutions of the stationary-phase cultures were prepared in TSB to inoculate the film-forming solutions. The initial cell concentration in the test solutions (≈107 CFU/mL) was determined by preparing appropriate decimal serial dilutions in the saline solution (0.85% *w*/*v* NaCl) of the TSB-diluted *E. coli* and *S. aureus* cell suspensions and plating them onto Mueller Hinton Agar (MHA) plates in triplicate, which were incubated at 37 °C for 24 h prior to enumeration (Stuart SC6 colony counter, Barloworld Scientific, Burlington, NJ, USA).

For the antimicrobial studies, 100 µL (33.3% (*v*/*v*)) or 250 µL (83.3% (*v*/*v*)) of the selected film-forming solutions was added to 1.5 mL Eppendorf tubes with, respectively, 200 or 50 µL of the corresponding TSB-diluted *E. coli* or *S. aureus* cell suspensions. Control samples were prepared by adding 100 or 250 μL TSB, respectively, to the bacterial suspensions. The samples were incubated at either 37 °C and 300 rpm for 24 h (VorTemp 56, Labnet International, Edison, NJ, USA) or 10 °C and 70 rpm for 5 or 10 days (Multitron Standard incubation shaker, Inforsh HT, Bottmingen, Switzerland) to simulate temperature abuse and refrigeration conditions. After the respective incubation periods, bacterial levels in test and control samples were determined by viable plate counting, as mentioned above. The initial pH and the pH after the respective incubation periods were measured in the test and control samples using a FiveEasy Plus pH-meter with the micro pH electrode LE422 (Mettler-Toledo, Greifensee, Switzerland).

### 3.9. Statistical Analysis

Statistical analysis was performed using SPSS software (version 26, IBM, Chicago, IL, USA) with t-test and one-way analysis of variance (one-way ANOVA). Tukey B was performed as a post hoc test when appropriate, and a significance level of *p* < 0.05 was used.

## 4. Conclusions

In this study, seaweed powder as a bio-filler was incorporated into alginate films with the aim of improving film properties. In addition, selected films were prepared with PAW due to its reported antimicrobial effects. Seaweed powder was incorporated at different concentrations (10, 30, and 50% (*w*/*w*)) and particle sizes (100 and 200 µm), which negatively affected the mechanical properties of the alginate film. The tensile strength and elongation at break further decreased with increasing filler concentration and bigger particle size (200 µm), demonstrating that the seaweed filler was poorly distributed within the alginate matrix. However, the water barrier and antioxidant properties significantly improved when seaweed filler was incorporated into films regardless of filler concentration and particle size. The mechanical properties (tensile strength) of alginate film prepared with PAW improved compared to the film prepared in distilled water, but a significant decrease was observed with seaweed filler, independently of its concentration. In addition, improved barrier properties were observed for the film prepared in PAW compared to the film prepared with distilled water, but preparation in PAW did not affect the antioxidant properties of the films. The antimicrobial activity of the alginate–seaweed film-forming solution was in general more pronounced when prepared with PAW and stored at 10 °C. A more pronounced inhibitory effect was observed on the Gram-positive *S. aureus* than on the Gram-negative *E. coli*, possibly because of differences in the cell wall and membrane of Gram-positive and Gram-negative bacteria. Additionally, a more pronounced antimicrobial activity was observed for the highest concentration of the film-forming solution (83.3% (*v*/*v*)) and at refrigerated temperature (10 °C).

Overall, this study has demonstrated that the addition of seaweed powder as a bio-filler, regardless of filler concentration and particle size, led to weaker films with enhanced barrier and antioxidant properties. The antioxidant and barrier properties were improved with increased filler loading, while the strength further decreased with increasing concentration and particle size, caused by the uneven distribution of filler within the alginate matrix. In addition, the study has demonstrated the potential of seaweed in combination with PAW towards enhanced functionality and bioactivity of alginate films for potential food packaging applications.

Disclaimer: The author, Estefanía Noriega Fernández, is employed with the European Food Safety Authority (EFSA) at the Nutrition and Food Innovation Unit that provides scientific and administrative support to the Panel on “Nutrition, Novel Foods and Food Allergens” in the area “Safety Assessment of Novel Foods”. However, the present article is published under the sole responsibility of the authors Hege Dysjaland, Izumi Sone, Estefanía Noriega Fernández, Morten Sivertsvik and Nusrat Sharmin and may not be considered as an EFSA scientific output. The positions and opinions presented in this article are those of the author/s alone and are not intended to represent the views/any official position or scientific works of EFSA. For more information about the views or scientific outputs of EFSA, please consult its website under http://efsa.europa.eu.

## Figures and Tables

**Figure 1 molecules-27-08356-f001:**
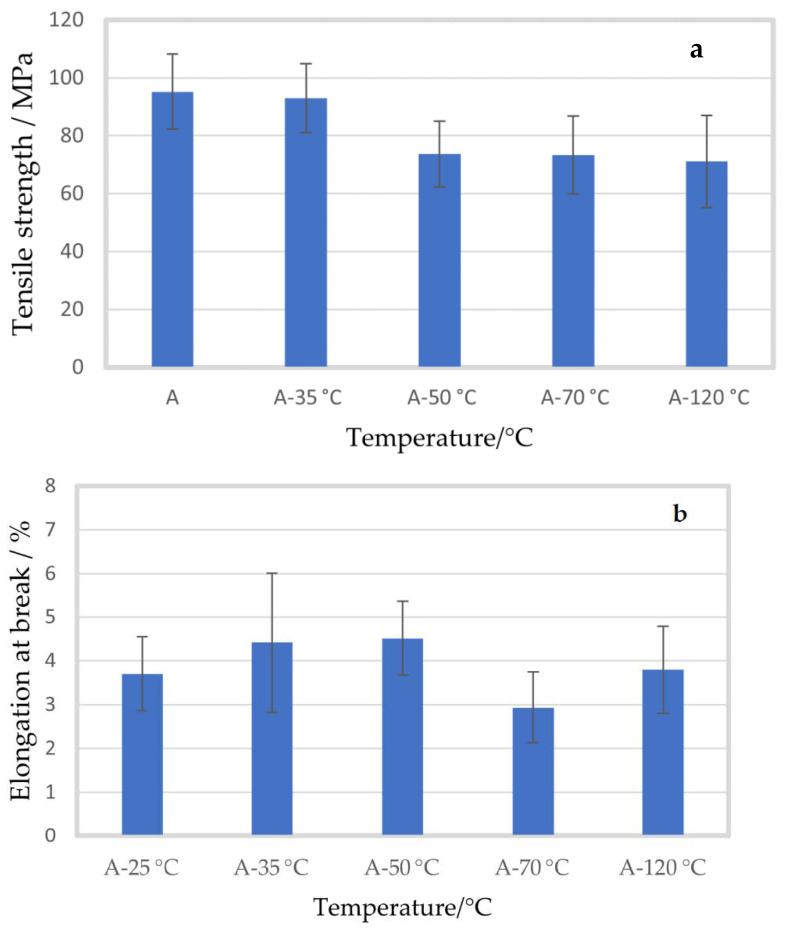
Mean tensile strength (**a**) and elongation at break (**b**) of alginate films (A) with standard deviation, prepared in distilled water at different temperatures.

**Figure 2 molecules-27-08356-f002:**
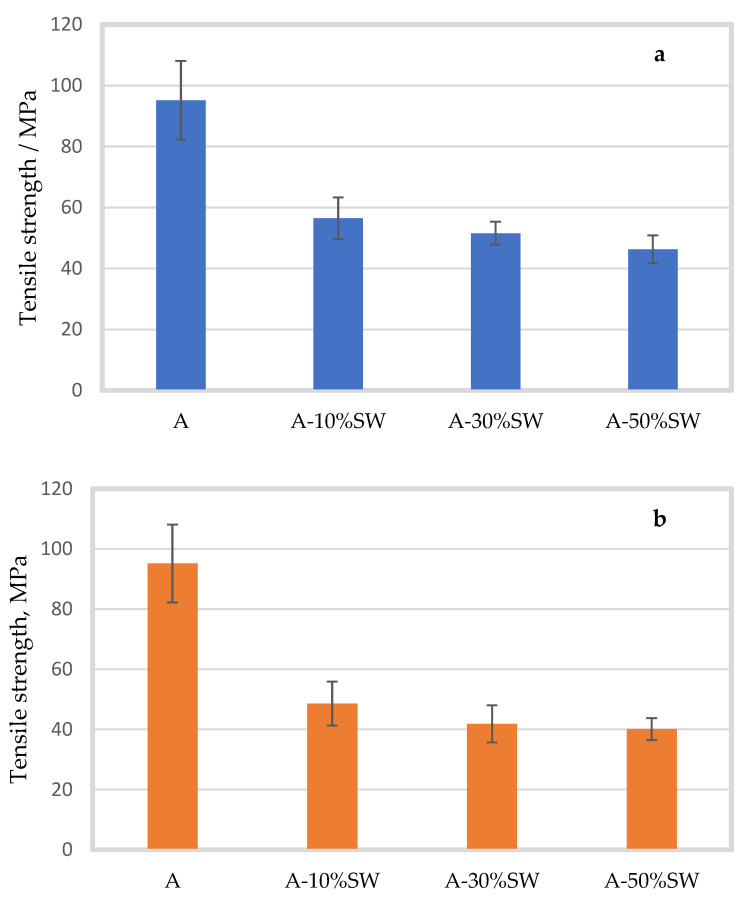
Mean tensile strength (MPa) of alginate films (A) with standard deviation, with seaweed (SW) filler of particle size 100 µm (**a**) (blue bars) and 200 µm (**b**) (orange bars) in concentrations of 10 (A-10%SW), 30 (A-30%SW) and 50% (A-50%SW) (*w*/*w*).

**Figure 3 molecules-27-08356-f003:**
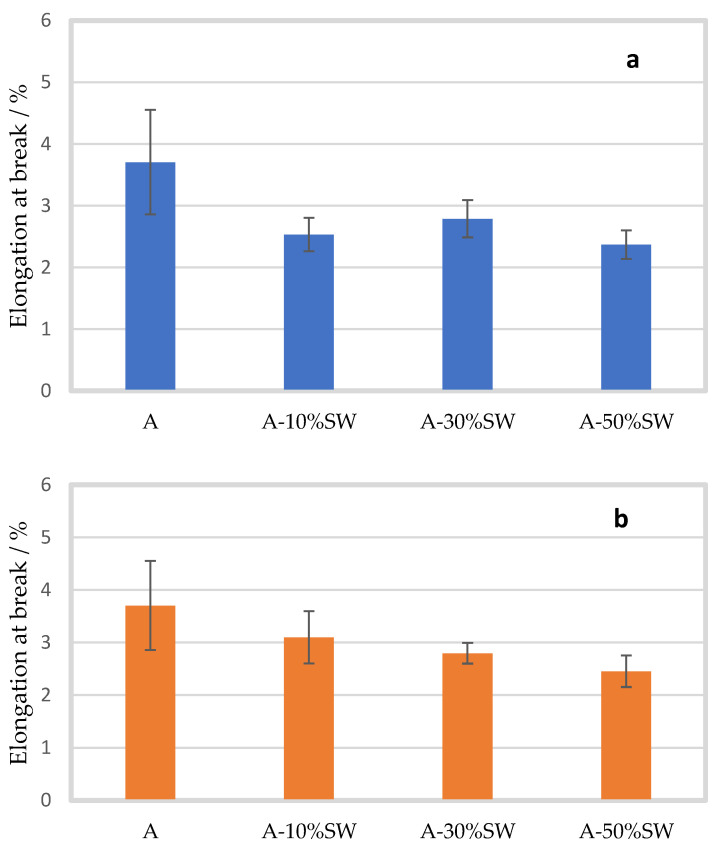
Mean elongation at break (%) of alginate films (A) with standard deviation, with seaweed (SW) filler of particle size 100 µm (**a**) (blue bars) and 200 µm (**b**) (orange bars) in concentrations of 10 (A-10%SW), 30 (A-30%SW) and 50% (A-50%SW) (*w*/*w*).

**Figure 4 molecules-27-08356-f004:**
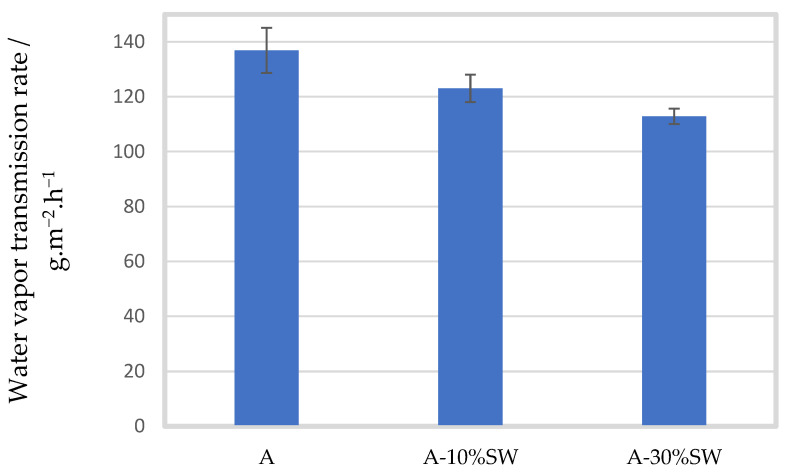
Mean water vapor transmission rate (g·m^−2^·h^−1^) of alginate films (A) with standard deviation, with seaweed filler (SW) of particle size 100 µm in concentrations of 10 (A-10%SW) and 30% (A-30%SW) (*w*/*w*).

**Figure 5 molecules-27-08356-f005:**
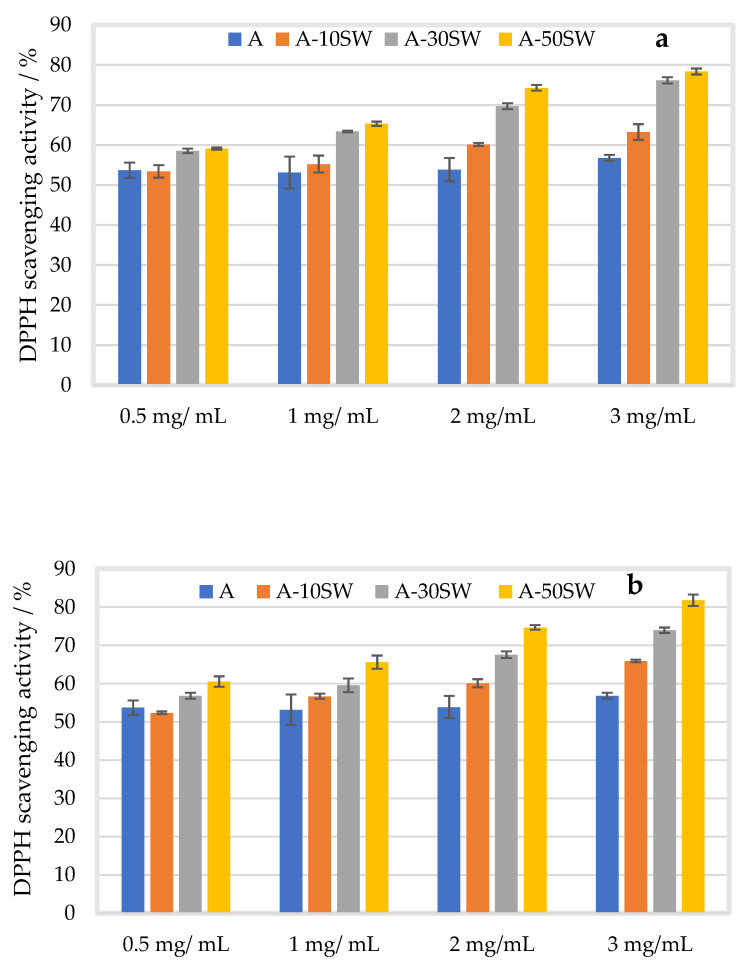
Mean DPPH scavenging activity (%) of alginate film (A, blue) with standard deviation at 0.5, 1.0, 2.0, and 3.0 mg/mL alginate, with and without filler (SW) of particle size 100 µm (**a**) and 200 µm (**b**) added at 10 (A-10%SW, orange), 30 (A-30%SW, grey) and 50% (A-50%SW, yellow) (*w*/*w*).

**Figure 6 molecules-27-08356-f006:**
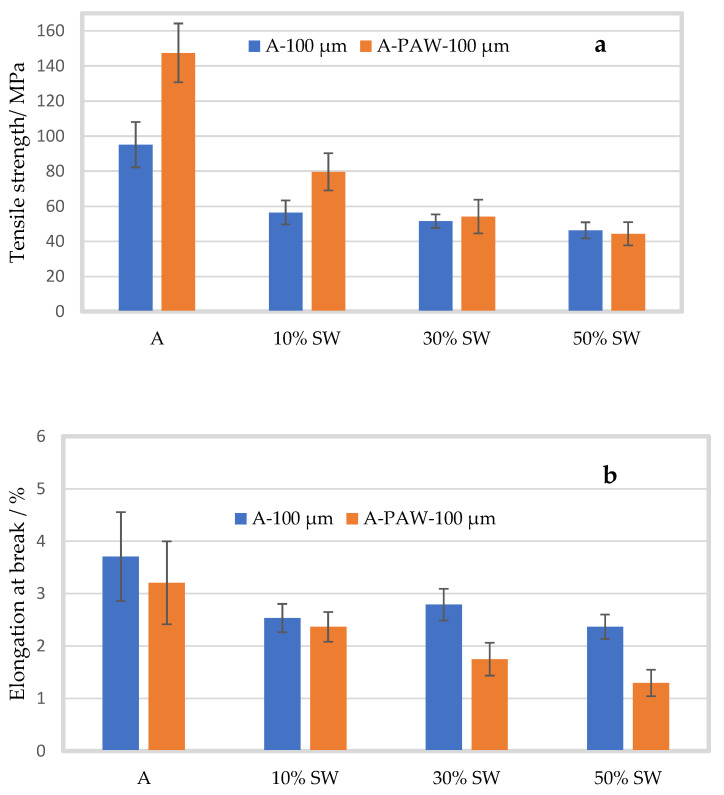
Mean tensile strength (**a**) and elongation at break (**b**) of alginate films prepared in distilled water (A, blue) and in PAW (A-PAW, orange) with standard deviation, with seaweed (SW) filler of particle size 100 µm in concentrations of 10 (A-10%SW), 30 (A-30%SW) and 50% (A-50%SW) (*w/w*).

**Figure 7 molecules-27-08356-f007:**
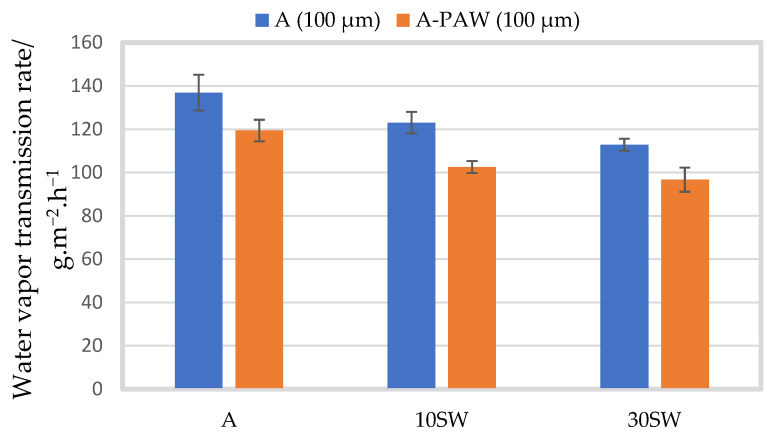
Mean water vapor transmission rate (g·m^−2^·h^−1^) of alginate films prepared in distilled water (A, blue) and in PAW (A-PAW, orange) with standard deviation, with seaweed (SW) filler of particle size 100 µm in concentrations of 10 (A-10%SW), 30 (A-30%SW) and 50% (A-50%SW) (*w*/*w*).

**Figure 8 molecules-27-08356-f008:**
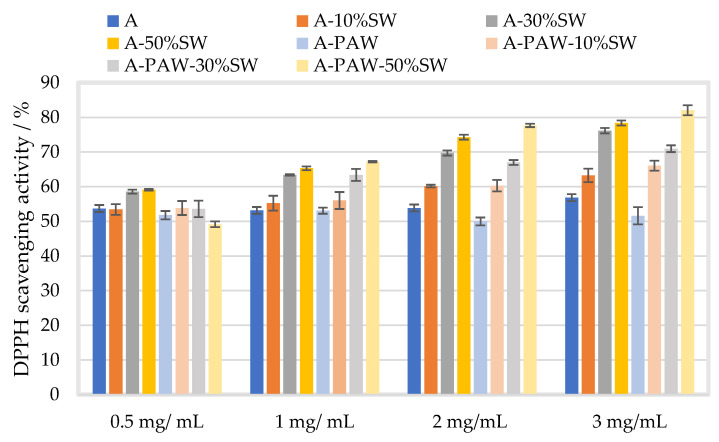
Mean DPPH scavenging activity (%) of alginate films prepared in distilled water (A, blue) and in PAW (A-PAW) with standard deviation, with seaweed (SW) filler of particle size 100 µm in concentrations of 10 (A-10%SW, orange), 30 (A-30%SW, grey) and 50% (A-50%SW, yellow) (*w*/*w*) in four different alginate concentrations (0.5, 1.0, 2.0, and 3.0 mg/mL). Blue bar represents films without filler.

**Figure 9 molecules-27-08356-f009:**
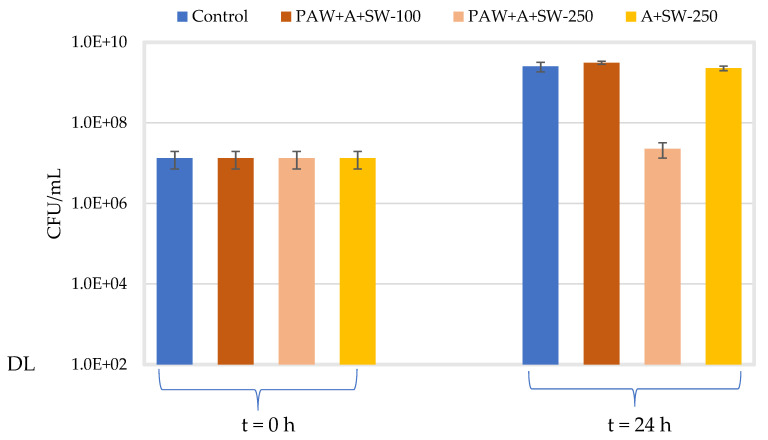
Viable counts (CFU/mL) of *E. coli* at t = 0 h for all the assayed conditions and after 24 h incubation at 37 °C and 300 rpm (blue: control in TSB; orange: alginate + PAW + 50% (*w*/*w*) seaweed–33.3% *v*/*v* (dark) or 83.3% *v*/*v* (light) of the film-forming solution; yellow: alginate + 50% (*w*/*w*) seaweed–83.3% *v/v* of film-forming solution). “DL” is the detection limit of the colony counting method (102 CFU/mL). Error bars represent the standard deviation (*n* ≥ 3).

**Figure 10 molecules-27-08356-f010:**
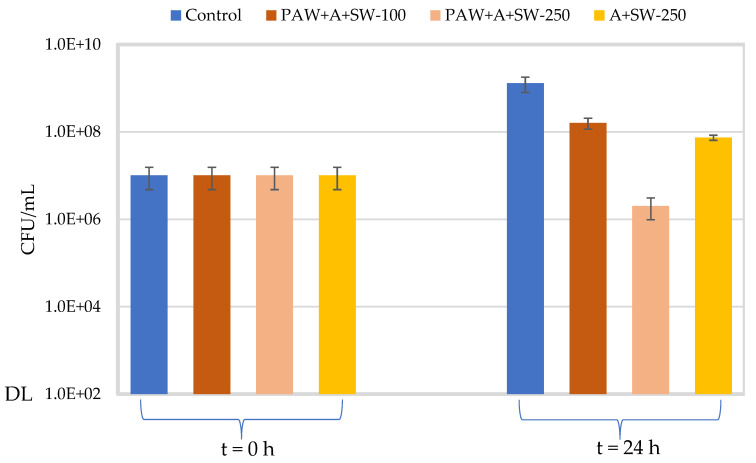
Viable counts (CFU/mL) of *S. aureus* at t = 0 h for all the assayed conditions and after 24 h incubation at 37 °C and 300 rpm (blue: control in TSB; orange: alginate + PAW + 50% (*w*/*w*) seaweed–33.3% *v*/*v* (dark) or 83.3% *v*/*v* (light) of the film-forming solution; yellow: alginate + 50% (*w*/*w*) seaweed–83.3% *v*/*v* of film-forming solution). “DL” is the detection limit of the colony counting method (102 CFU/mL). Error bars represent the standard deviation (*n* ≥ 3).

**Figure 11 molecules-27-08356-f011:**
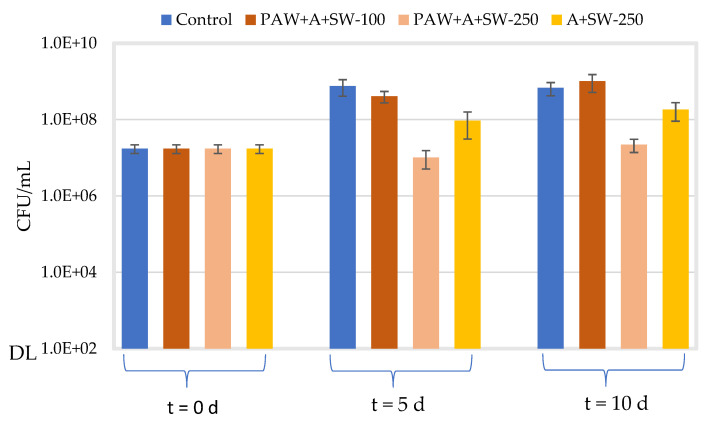
Viable counts (CFU/mL) of *E. coli* at t = 0 d for all the assayed conditions and after 5- and 10-day incubation at 10 °C and 70 rpm (blue: control in TSB; orange: alginate + PAW + 50% (*w*/*w*) seaweed–33.3% *v*/*v* (dark) or 83.3% *v*/*v* (light) of the film-forming solution; yellow: alginate + 50% (*w*/*w*) seaweed–83.3% *v*/*v* of film-forming solution). “DL” is the detection limit of the colony counting method (102 CFU/mL). Error bars represent the standard deviation (*n* ≥ 3).

**Figure 12 molecules-27-08356-f012:**
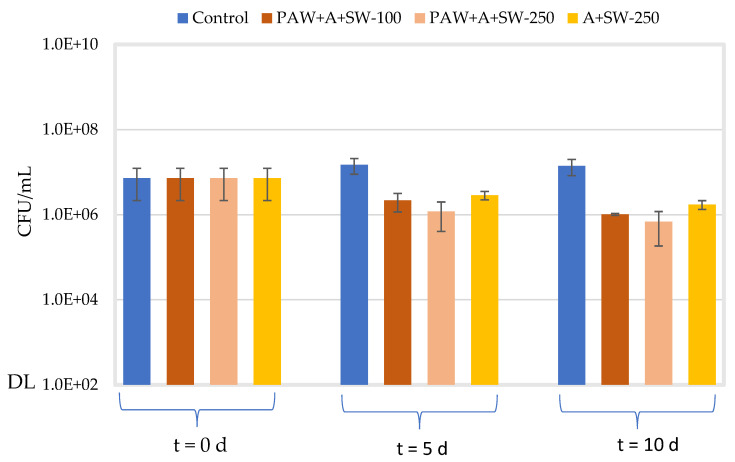
Viable counts (CFU/mL) of *S. aureus* at t = 0 d for all the assayed conditions and after 5- and 10-day incubation at 10 °C and 70 rpm (blue: control in TSB; orange: alginate + PAW + 50% (*w*/*w*) seaweed–33.3% *v*/*v* (dark) or 83.3% *v*/*v* (light) of the film-forming solution; yellow: alginate + 50% (*w*/*w*) seaweed–83.3% *v*/*v* of film-forming solution). “DL” is the detection limit of the colony counting method (102 CFU/mL). Error bars represent the standard deviation (*n* ≥ 3).

## Data Availability

Not applicable.

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
