# Peer review of "Mechanical, Barrier, Antioxidant and Antimicrobial Properties of Alginate Films: Effect of Seaweed Powder and Plasma-Activated Water"

_molecules, 2022, doi:10.3390/molecules27238356_

Round 1

Reviewer 1 Report

This manuscript describes that enhancing the properties of biopolymer films by alginate and crushed seaweed. Representing data is reliable and these results are interesting for readers of this journal. Therefore, I recommend accepting  after minor revised.

My major suggestion is below:

Please describe the M/G ratio of the alginate used and information on dyads and triads if possible. As also described by the authors, MG ratios and their sequences contribute to the physicochemical properties of alginate. Therefore, in order to ensure the reproducibility of the experiment, it is helpful for the reader to describe this information as much as possible.

In addition, please provide specific information about the ingredients of the milled seaweed used. I have no doubt that this addition contributes to the improvement, but if possible, please mention which ingredients are the main ones and which factors have a positive effect. If its factor is considered as alginate, it is desirable to present the same information as for commercial alginate above.

Other minor concerns are:

Line 18: Insert a space before µm.

Line 32: E. coli should be written as Escherichia coli (in italic).

Other scientific names should also be italicized in this manuscript.

Line 60: Please mention MG block.

Line 130: C3H8O3, these numbers should be subscripted.

Line 609: 6 of 109 should be superscripted. Please check other lines in this manuscript by yourself.

Figs. 5 and other applicable figures: No legends should be shown below the x-axis.

Author Response

This manuscript describes that enhancing the properties of biopolymer films by alginate and crushed seaweed. Representing data is reliable and these results are interesting for readers of this journal. Therefore, I recommend accepting  after minor revised.

 My major suggestion is below:

Please describe the M/G ratio of the alginate used and information on dyads and triads if possible. As also described by the authors, MG ratios and their sequences contribute to the physicochemical properties of alginate. Therefore, in order to ensure the reproducibility of the experiment, it is helpful for the reader to describe this information as much as possible.

Response: The M/G ratio: 2.12 has been added to the manuscript

In addition, please provide specific information about the ingredients of the milled seaweed used. I have no doubt that this addition contributes to the improvement, but if possible, please mention which ingredients are the main ones and which factors have a positive effect. If its factor is considered as alginate, it is desirable to present the same information as for commercial alginate above.

Response: Unfortunately, no compositional analysis was conducted on the seaweed powder. Therefore, we are unable to add the main ingredients of the seaweed powder in use.

 Other minor concerns are:

Line 18: Insert a space before µm.

Response: Added

Line 32: E. coli should be written as Escherichia coli (in italic).

Response: Modified

Other scientific names should also be italicized in this manuscript.

Line 60: Please mention MG block.

Response: The MG blocks are already mentioned in the text.

Line 130: C3H8O3, these numbers should be subscripted.

Response: The subscription has been added to the manuscript.

Line 609: 6 of 109 should be superscripted. Please check other lines in this manuscript by yourself.

Response: Checked

Figs. 5 and other applicable figures: No legends should be shown below the x-axis.

Response: The legends have been moved (please check the manuscript)

Reviewer 2 Report

Dear Authors,

The manuscript is high quality and I only have some minor comments:

1: I can se no levels of significance indicated on any of the figures. Please indicated the different statistical relationships between the presented data sets on all figures.
2: I suggest removing data sets from figures, if they were not actually measured (like barrier properties - 50% SW).

3: Figures regarding the antimicrobial tests are very confusing right now. Please rescale the y axis, as it can be seen that several samples had no initial CFU counts.

Author Response

Dear Authors,

The manuscript is high quality and I only have some minor comments:

1: I can se no levels of significance indicated on any of the figures. Please indicated the different statistical relationships between the presented data sets on all figures.

Response: The level of significance was clearly mentioned in the text. Adding the significance level to the figure would make the figures look complicated. That’s why the significance level was not added to the figures.

2: I suggest removing data sets from figures, if they were not actually measured (like barrier properties - 50% SW).

Response: Removed

3: Figures regarding the antimicrobial tests are very confusing right now. Please rescale the y axis, as it can be seen that several samples had no initial CFU counts.

Response: The y-axis was set in such a way that the detection limit could be mentioned and the authors would like to keep it that way.